# Effects of Physical Exercises Alone on the Functional Capacity of Individuals with Obesity and Knee Osteoarthritis: A Systematic Review

**DOI:** 10.3390/biology11101391

**Published:** 2022-09-23

**Authors:** Vanessa Silva Caiado, Aline Cristina Gomes Santos, Eloá Moreira-Marconi, Marcia Cristina Moura-Fernandes, Adérito Seixas, Redha Taiar, Ana Cristina Rodrigues Lacerda, Anelise Sonza, Vanessa Amaral Mendonça, Danúbia Cunha Sá-Caputo, Mario Bernardo-Filho

**Affiliations:** 1Programa de Pós-Graduação em Fisiopatologia Clínica e Experimental, Universidade do Estado do Rio de Janeiro, Rio de Janeiro 20551-030, Brazil; 2Laboratório de Vibrações Mecânicas e Práticas Integrativas-LAVIMPI, Departamento de Biofísica e Biometria, Instituto de Biologia Roberto Alcântara Gomes, Policlínica Piquet Carneiro, Universidade do Estado do Rio de Janeiro, Rio de Janeiro 20950-003, Brazil; 3Programa de Pós-Graduação em Ciências Médicas, Universidade do Estado do Rio de Janeiro, Rio de Janeiro 20551-030, Brazil; 4Escola Superior de Saúde, Universidade Fernando Pessoa, 4249-004 Porto, Portugal; 5MATériaux et Ingénierie Mécanique (MATIM), Université de Reims Champagne-Ardenne, 51100 Reims, France; 6Faculdade de Ciências Biológicas e da Saúde, Universidade Federal dos Vales do Jequitinhonha e Mucuri, Diamantina 39100-000, Brazil; 7Programa de Pós-Graduação em Fisioterapia, Programa de Pós-Graduação em Ciências do Movimento Humano, Residência Multiprofissional em Saúde da Família e Comunidade, Universidade do Estado de Santa Catarina, Florianópolis 88035-901, Brazil; 8Departamento de Fisioterapia, Faculdade Bezerra de Araújo, Rio de Janeiro 23052-180, Brazil

**Keywords:** physical function, knee osteoarthritis, obesity, physical exercise alone, rehabilitation

## Abstract

**Simple Summary:**

Osteoarthritis is a degenerative joint disease that affects millions of people around the world. Knee osteoarthritis is one of the causes of more significant functional disability among people with it. Currently, obesity is identified as one of the main risk factors for the onset of the disease due to excess load on the joints of the lower limbs, especially the knees. The association of measures, such as weight reduction through diets and exercise, can alleviate symptoms and increase the physical condition of people affected by these clinical conditions. However, many individuals with obesity have difficulty adhering to diet programs and need to improve in order to perform their functional activities. The aim of this systematic review was to evaluate the results of several physical exercise programs conducted without the association of diet, demonstrating the improvement of the functional capacity of individuals with these concomitant clinical conditions, presenting another proposal to reduce the symptoms of the disease in this population.

**Abstract:**

The association between obesity and knee osteoarthritis (KOA) is reported in the literature. The inflammatory factors described in obesity associated with mechanical overload on the knee joint lead to KOA development and reduced functional capacity in these individuals. Most physical exercise (PE) protocols associate a diet program to improve the functional capacity of individuals with concomitant KOA and obesity. There is a lack of published protocols performing PE alone, which would be without an associated diet program in individuals with both clinical conditions. In this systematic review, the authors summarize the effects of the application of PE alone, describing each protocol and reporting the improvement in the function of people with these clinical conditions. This investigation was conducted according to the PRISMA guidelines and registered in PROSPERO. Five databases (MEDLINE/PubMed, PEDro, Scopus, CINAHL and Web of Science) were used up to July 2022 and ten studies, including 534 participants, met the inclusion criteria. The PEDro scale, Cochrane collaborations and ROBINS-I tools were used to evaluate the methodological quality and risk of bias. It was concluded that PE performed alone seems to provide an improvement in the functional capacity of these individuals even without an associated diet plan in the condition of obesity.

## 1. Introduction

According to the Osteoarthritis Research Society International (OARSI), osteoarthritis (OA) is a disorder involving mobile joints [1]. It is characterized by cellular stress and the degradation of the extracellular matrix initiated by micro- and macro-lesions that activate maladaptive repair responses [1]. The mechanism involved in OA includes pro-inflammatory pathways of innate immunity followed by anatomical and/or physiological disorders characterized by cartilage degradation, bone remodeling, formation of osteophytes, joint inflammation and loss of normal joint function [1]. Knee OA (KOA) is considered a major health care burden throughout the world responsible for generating significant pain and disability [2].

Some comorbidities associated with KOA, such as obesity, have been appointed as a modifiable risk factor of the development or worsening of symptoms in KOA [3]. Two in three people with obesity are at a risk of developing symptomatic KOA in at least one knee [4]. The World Health Organization defines obesity as a condition in which a person has a body mass index (BMI) of 30 kg/m^2^ or more and includes overweight people if their BMI is equal to or more than 25 kg/m^2^ [5]. The increased mechanical overload in the knee joint, in addition to muscle weakness of the lower limbs due to obesity, causes an exacerbation of pain, aggravating the disease and generating a decrease in functional capacity, followed by consequent inactivity [6]. Ventura et al. [7] pointed out that functional capacity can be defined as the abilities that are essential for an individual to function independently in a variety of community settings, including work and social situations. Commonly, some questionnaires and functional tests can assess lower limb muscle strength or the performance of activities of daily living such as walking distance, sitting and rising from a chair and climbing stairs [8]. Gomes-Neto describes that people with overweight, obesity and KOA, due to pain and poor adherence to physical activity, can present with difficulty in performing daily living activities, a slow gait speed and poor lower limb muscle strength, showing a reduced functional capacity [9].

Protocols of physical exercise (PE) are considered an option for the non-pharmacological treatment of OA [10]. Moreover, the American College of Rheumatology (ACR) strongly recommends PE and weight loss to patients with OA who present with obesity or overweight [11]. But there is a lack of studies that report on the effects of PE alone (not including or associated with diet) on the functional capacity in this population.

Lee et al., 2013 [12] stated that obesity or overweight is significantly associated with physical inactivity in adults with KOA, leading to increased health care costs. PE has been recommended to interrupt the cycle of obesity−KOA−pain−inactivity [6]. It can improve functional capacity due to the increase the muscle strength, the alleviation of pain and the modification of joint biomechanics [3]. These factors result in a decrease in joint overloading, playing an important role in the prevention or delay of disability in this population [13]. Therefore, the aim of this systematic review was to summarize the effects of various modalities of PE alone on the functional capacity of individuals with overweight and/or obesity in association with KOA.

## 2. Material and Methods

### 2.1. Protocol and Registration

This systematic review follows the Preferred Reporting Items for Systematic Reviews and Meta-Analysis (PRISMA) and it was registered on the international prospective registry of systematic reviews (PROSPERO) number CRD42020196936 [14].

### 2.2. Research Question

This systematic review proposes to answer the following question: What are the effects of physical exercise alone on the functional capacity of individuals with obesity and overweight and knee osteoarthritis? The PICO strategy was used to define the components of the research question: Participants (P) = individuals with overweight, obesity and KOA; Interventions (I) = physical exercise; Comparators (C) = control group with no exercise practice or different types of physical exercise alone; Outcome (O) = functional capacity.

### 2.3. Inclusion and Exclusion Criteria

The inclusion criteria were articles reporting original data on the functional capacity of individuals with obesity, overweight and KOA. The studies include: (i) investigations about the effects of PE alone (not included diet protocols) on functional capacity considering gait speed, specific questionnaires about daily living performance, specific functional tests or quantitative measures of the muscle strength of the lower limbs of individuals with obesity, overweight and KOA; (ii) interventionist studies with inter- or intra- group comparisons; (iii) studies published in the English language; and (iv) patients that performed static or dynamic exercises. As exclusion criteria, (i) review articles; (ii) case reports; (iii) studies with animals; (iv) studies involving surgery or pharmacology; (v) studies related to other diseases or samples of people without obesity and (vi) exercises associated with other therapies, such as diet or other interventions, were eliminated.

### 2.4. Search Strategies

Five electronics databases: MEDLINE/Pubmed; Physiotherapy Evidence Database (PEDro); Scopus; Cumulative Index to Nursing and Allied Health Literature (CINAHL) and Web of Science were searched by three reviewers independently until July 2022. The string for the search strategy is shown in the Appendix A.

A.C.G.S. and E.M.-M. independently screened the titles and abstracts according to the inclusion criteria. In the case that the eligibility was uncertain based on the title and abstract, the full text of the study was obtained.

### 2.5. Study Selection

All publications found on the databases were exported to an Excel spreadsheet, and the duplicates were manually removed by two authors (A.C.G.S. and E.M.-M.). Two reviewers (A.C.G.S. and E.M.-M.) independently applied the eligibility criteria and selected the studies for inclusion in the systematic review (researchers were blinded to each other’s decisions). Disagreements were solved by the analysis of a third author (V.S.C.).

The data were extracted from each article and imported to an Excel spreadsheet containing: (i) data regarding study information (author and year), (ii) aim, (iii) participants/groups (sample size, age, sex), (iv) body mass index, (v) physical exercise programs, (vi) functional capacity assessment, (vii) methodological quality (PEDro scale) and (viii) functional capacity outcomes. Two researchers (A.C.G.S. and E.M.-M.) independently performed the data extraction. Disagreements were resolved by a third author (V.S.C.).

### 2.6. Methodological Quality, Risk of Bias and Levels of Evidence (LE) of the Selected Papers

The studies were independently appraised by two reviewers (V.S.C. and A.C.G.S.) and if there were any disagreement, a third reviewer (E.M.-M.) was consulted. The issue was discussed until a consensus was reached. The methodological quality was evaluated according to the PEDro scale, which consists of a checklist with ten items established based on an “expert consensus”, and this scale is specific to clinical trials of physical therapy interventions [15]. The publications were classified as ‘high’ methodological quality (score of seven or greater), ‘fair’ methodological quality (score of five to six) and ‘poor’ methodological quality (score of four or below) [15].

The risk of bias in the included studies were evaluated by two reviewers (V.S.C. and A.C.G.S.) and if there were any disagreement, a third reviewer (E.M.-M.) was consulted. The risk of bias in the randomized controlled trials (RCT) was determined using the Cochrane Collaboration’s tool [16]. Each domain was qualified as having a low risk, unclear risk, or high risk of bias. Each judgment was represented by the colors green, yellow and red, respectively. The risk of bias of the other studies was determined using the Risk of Bias In Non-randomized Studies of Interventions tool (ROBINS-I) [17] and each domain was qualified as low risk, moderate risk, serious risk, critical risk of bias or no information to judge this domain. Each judgment was represented by the colors green, yellow, red, dark red and blue, respectively.

### 2.7. Data Synthesis

Due to the different protocols and outcome measures found in the included studies, statistical pooling of the data was not appropriate.

## 3. Results

A total of 674 papers were initially screened (PubMed = 107, PEDro = 51, Scopus = 204, CINAHL = 89, Web of Science = 223). After removing duplicates, a total of 367 records remained and 10 articles fulfilled the inclusion criteria. The selection process is schematized in the PRISMA flowchart [14] (Figure 1).

Table 1 summarizes the publications selected in this systematic review, presenting the aims, characteristics of the participants, programs of PE alone, methodological quality with score on the PEDro scale and the outcomes related to the effects of the physical exercises alone on functional capacity in individuals with overweight, obesity and KOA.

### 3.1. Study Population

The selected studies included a total of 534 individuals, including people with obesity and/or overweight who had a diagnosis of unilateral or bilateral KOA. Regarding the inclusion criteria of the BMI, two studies [18,19] analyzed only women with obesity (BMI ≥ 30 kg/m^2^) [5]. One study [13] investigated only men with obesity.

#### Interventions

The duration of treatment ranged from 4 to 24 weeks with the number of sessions ranging from 8 to 144 sessions. The selected studies carried out different interventions based on isometric exercises, aerobic training or a combination of multiple forms of exercise: (a) two [19,20] investigated the effects of aquatic exercises, (b) three [21,22,23] combined isometric exercises with aerobic training, (c) one [24] applied only a treadmill walking program, (d) one [25] compared an underwater treadmill exercise with home exercises, (e) one [13] analyzed just isometric exercises, (f) one [18] compared a program of weight bearing (WB) with non-weight bearing (NWB) exercises for muscle strengthening and (g) one reported the short-term effects of a home protocol of NWB exercises [26].

### 3.2. Functional Capacity Assessments

The quantitative results found on functional capacity are shown in the table with the statistical values mentioned in the studies.

Nine varied tools were used to compare the measures obtained before and after the protocols utilized in the studies. The functional capacity was evaluated through the improvement in muscle strength and in some daily activities, and it was assessed through specific questionnaires associated with these daily activities [9]. In addition, the assessment of functional activities can confirm improvements after the protocols used in the studies [27]. The functional tests used to evaluate these activities were: the Short Physical Performance Battery (SPPB) [27], the 6-min walk test [25], 40-m fast-paced walk test expressed as speed in meters/second, 30-s chair sit-to-stand test [18], 6-step stair-climb and descent test [18], walking speed and standing balance [23].

Some specific questionnaires related to the assessment of pain and the function of the knee joint, specifically related to KOA, are widely used all over the world [13]. The Western Ontario and McMaster Universities Osteoarthritis Index (WOMAC) is recommended by the ACR to assess KOA patients [13,18,19,20,21,22,23]. It is a self-administered questionnaire with three domains which evaluate pain, joint stiffness and functional capacity [13]. The Knee Injury and Osteoarthritis Outcome Score (KOOS) questionnaire is a standardized and self-administered questionnaire commonly used by researchers to evaluate pain, knee join function and other associated problems [18,24]. The KOOS assessment uses five categories, which are: (i) Pain; (ii) Symptoms; (iii) Activities of daily living; (iv) Function, Sports and Recreational Activities; and (v) Quality of life. The Canadian Occupational Performance Measure (COPM) is designated to verify changes in occupational performance over time in activities that the individual has self-identified as difficult to perform [24].

### 3.3. Methodological Quality

Considering the methodological quality of randomized clinical trials in physiotherapy, assessed by the PEDro scale, seven studies [13,18,19,20,22,25,26] were considered to be ‘high’ methodological quality (≥7) and one [23] was ‘fair’ methodological quality (5 or 6). Two studies [21,24] could not be evaluated by the PEDro scale as they were not RCTs.

### 3.4. Risk of Bias

A detailed assessment of the risk of bias of the RCT, carried out according to the Cochrane Collaboration’s tool, is presented in Figure 2A. Five publications [18,19,22,25,26] were classified as having a low risk of bias, one publication [13] was classified as having an uncertain risk of bias and two publications [20,23] were classified as having a serious risk of bias. The assessment for the non-RCT, carried out using the ROBINS-I tool [17], is presented in Figure 2B. One publication [24] was classified as having a low risk of bias, despite being a non-RCT and one publication [21] was classified as having a serious risk of bias.

## 4. Discussion

Physical exercise performed alone, regardless of the modality, improved the functional capacity of individuals with obesity, overweight and KOA. Most studies reported statistically significant values (*p* ≤ 0.05) regarding the improvement in functional capacity.

People with overweight or obesity can have their functional capacity affected by the physical inactivity, muscle weakness and pain associated with KOA [3]. The American College of Sports and Medicine (ACSM) Guidelines [28] report that resistance training can reduce pain and disability in individuals with osteoarthritis. Impaired muscle function is associated with knee pain and physical inactivity, and they are commonly seen in individuals with obesity, mainly if they are included in the KOA population [3].

A relevant issue in the management of KOA individuals with obesity is the recommendation of the Ottawa Panel Evidence-based Clinical Practice (OPECP) Guideline [29] about reducing weight prior to the implementation of WB exercises to avoid dysfunction and maintain the joint integrity. Despite that, Inam et al., 2020 [30] stated that a low number of patients receive weight loss concealing by their physicians. The association between diet and exercise may be related to the best overall improvements in self-reported measures of functional capacity, pain and mobility in this population when properly oriented [31]. PE is recommended to interrupt the cycle of obesity−KOA−pain−inactivity [6]. The best type of PE and the effects when performed alone in this specific population affected by KOA and obesity is not yet clear in the literature [18]. There is some evidence suggesting that low-impact PE may be more effective than high-impact exercise for this population [18]. Despite this, some studies have shown beneficial effects on the functional capacity of individuals with obesity, overweight and KOA after performing low- or high-impact exercises alone or supplemented by aerobic training [13,18,32].

OARSI Guidelines [33] for the non-surgical management of KOA recommend land-based exercises as appropriate. The land-based exercises can be described as one or more exercises including a combination of strength training, active range of motion exercises and aerobic activity [18]. In this program modality, both WB exercises and NWB exercises are included. Bennel et al., 2020 [18] compared home-based NWB quadriceps strengthening exercises with WB exercises for 12 weeks in women with KOA and obesity. Both groups reported improvements in pain and function, although the WB exercises obtained better results for function. Regarding NWB exercises, Rafiq et al., 2021 [26] reported short-term effects on the functional capacity of populations with obesity, overweight and KOA after 4 weeks of NWB exercises at home, concluding that there was no improvement in the physical function scores of WOMAC [26]. Messier et al., 2010 [31], compared six months of WB exercise alone with an exercise−diet blended program in older adults with KOA and obesity, reporting improvements in walking velocity through kinematic analysis and muscular strength by isokinetic dynamometer in both groups. This shows that weight bearing exercises, which are considered high impact, could also provide benefits for people with obesity and KOA.

Aquatic exercises are also recommended as an intervention for individuals with osteoarthritis, including KOA [10,11]. Aquatic exercises would reduce the overcharge on the knee joint, allowing for improvements on tolerance levels to support the exercises. Casilda-López et al., 2017 [19] assessed the functional capacity of women with obesity and KOA after 8 weeks of aquatic dance-based exercises 3 times per week versus a control group which performed conventional aquatic exercises. Both groups showed improvements in the physical function domain of the WOMAC questionnaire, but displayed different *p* values between the groups, with there being a more significant difference in the experimental group. Taglietti et al., 2018 [34] compared aquatic exercises alone with patient education in individuals with KOA during an eight-week intervention. They stated the improvement in functional capacity through reduced values on the WOMAC scale in the aquatic exercise intervention group. Wang et al., 2010 [35] compared aquatic with land-based exercise versus a control group over 12 weeks. They concluded that both aquatic and land-based exercise achieved positive results on the Knee Injury and Osteoarthritis Outcome Score and six- minute walk test. Lim et al., 2010 [20] investigated 3 groups according to different modalities of PE alone: an aquatic group performed a protocol including strength training and aerobic training using underwater bicycling; the land-based group performed joint mobilization, stretching and strengthening exercises; while the control group maintained home –based exercises. Both exercise groups showed a significant improvement in lower limb function when measured by the WOMAC, compared with the control group. Kuptinaratsaikul et al., 2019 [25] investigated the effects of an underwater treadmill (UTM) exercise regimen compared to home exercise regarding the functional capacity of individuals with KOA and obesity. The study group underwent treadmill exercise for 30 min/day, three times/week for four weeks and the control group performed daily quadricep exercises at home for 30 min at the same frequency. They reported significant improvement in the two groups in the 6-min walking distance and quadricep strength at the end of the study. UTM and home-based quadricep exercise were equally effective.

Another type of PE that would reduce the overload on the knee joint is a technology called lower body positive pressure (LBPP), which consists of a treadmill using a waist-high air chamber filled with positive air pressure, used to accurately and reliably diminish body weight during PE, allowing for a longer exercise time walking on the treadmill for people with overweight and obesity and KOA [36]. Mainly for patients at a risk of the exacerbation of knee joint symptoms following PE, LBPP can be a safe way to perform WB exercises [36]. Liang et al., 2019 [37] relayed the improvement in WOMAC scores after using a LBPP protocol six times per week for 30 min/day for two weeks as an intervention for patients with KOA. Peeler et al., 2018 [24] examined the effect of a 12-week LBPP on joint function, thigh strength and the ability to perform normal activities of daily living in a single group with obesity, overweight and KOA. Significant increases in strength were observed between baseline and follow-up testing at all three isokinetic testing speeds. Takacs et al., 2013 [38] investigated the functional status of overweight individuals with KOA between two 20-min treadmill walking sessions. There were no changes in KOOS but a feeling of safety was relayed by the participants of the study.

Another way to obtain gains in functional capacity in individuals with KOA is quadricep strengthening alone through isometric or concentric training. Jenkinson et al., 2009 [39] compared after six months four groups of individuals with KOA and obesity randomized into three protocols of diet alone, diet plus exercise and quadricep strengthening alone versus a control group. They reported a significant reduction in the WOMAC physical function score in groups that performed quadriceps strengthening alone. Mahmoud et al., 2017 [13] evaluated the effects of isometric exercise training on quadricep strength versus a control group in males with KOA and obesity three times a week for 12 weeks. The maximal voluntary isometric knee extension torque was measured using an isokinetic dynamometer and the functional capacity was evaluated using the WOMAC. All the domains in the WOMAC, including the functional capacity, were significantly improved in both groups but in the isometric group, it was significantly better. Huang et al., 2017 [40] showed a relief in joint pain and an improvement in the functional capacity of individuals with KOA after performing a quadricep isometric contraction exercise protocol.

OPECP Guidelines for the management of osteoarthritis in adults with obesity or overweight [29] stated in its publication a Grade A (strongly recommended) recommendation for PE including strength training and aerobic training. People with KOA and the comorbidity of obesity have an increase in the amount of fat surrounding the quadricep muscles, which may increase disability due to a decrease in lower extremity performance [29]. Semanik et al., 2012 [41] reported that aerobic training and strengthening can improve function and health status among patients with KOA. Schlenk et al., 2011 [23] investigated the effects of a 6-month strength training and a fitness walking program 150 min/week supplemented with lower extremity exercise monitored over the phone. The functional capacity of individuals with KOA and obesity was evaluated by SPPB, the 6-min walking test and the WOMAC questionnaire at the end of study. There were increases in the SPPB scores, no significant differences in the WOMAC scores, but a greater improvement in the mean distance of the 6-min walk. Mihalko et al., 2019 [42] found an improvement in gait efficacy and balance efficacy in people with KOA and obesity after 6 months of combined walking with strengthening. Oyeyemi et al., 2013 [21] evaluated people with KOA and obesity after a program of isometric quadricep exercise and bicycle ergometry with a graduated increase in intensity. Overall, the program had significant post treatment benefits on function, assessed by the WOMAC questionnaire. Krasilshchikov et al., 2011 [22] reported significant outcomes in physical function scores, knee extensor concentric peak torque at 120^◦^ and 180^◦^/seconds and six-minute walking distance after a protocol combined resistance and aerobic training. A program of PE performed alone can generate different types of strengthening [43]. Kabiri et al., 2018 [32] reported that in KOA patients treated with several modalities of aerobic training combined with the strengthening of lower limbs, reduction in pain and improvements in functional capacity were observed.

## 5. Limitations

A lack of published studies meeting the inclusion criteria of this review could have limited a consensus on the effects of PE performed alone for the improvement of functional capacity in individuals with obesity and KOA. Moreover, only papers published in English into well-known databases were considered and, therefore, important publications in other databases may not have been accessed. Furthermore, the included studies may have had some limitations such as small sample sizes and heterogeneous samples.

## 6. Conclusions

Despite the limited number of studies meeting the inclusion criteria of this review, PE performed alone without associated diet programs seems improve the functional capacity of individuals with obesity and KOA. The studies in this review showed benefits with programs containing different types of PE. The PEs performed alone were aerobic and isometric training, aquatic exercises, quadricep strengthening and WB and NWB exercises. Although some guidelines recommend non-weight bearing exercises for people with obesity to avoid the impact on the joints, weight bearing and other types of muscle strengthening contributed to lower limb strengthening.

Considering the comorbidity of obesity in the population with KOA and the low adherence to associated diet programs for weight loss, knowing that the performance of exercises alone brings benefits for functionality can offer another path of guidance as a tool to approach these individuals.

## Figures and Tables

**Figure 1 biology-11-01391-f001:**
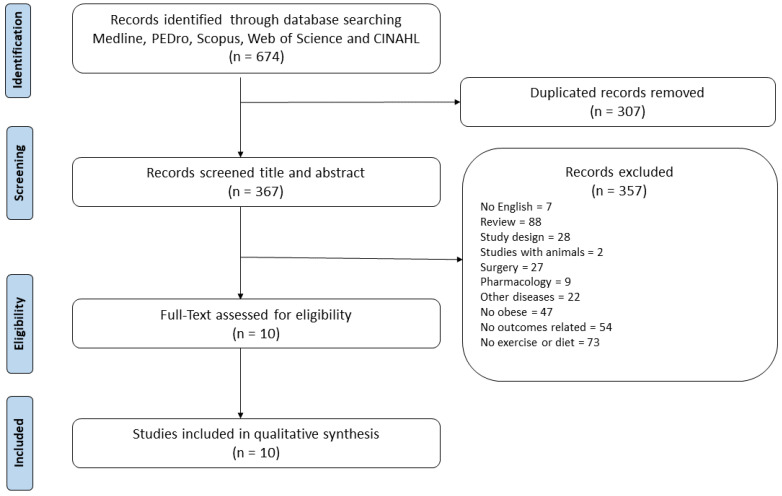
PRISMA flow diagram of the literature selection process.

**Figure 2 biology-11-01391-f002:**
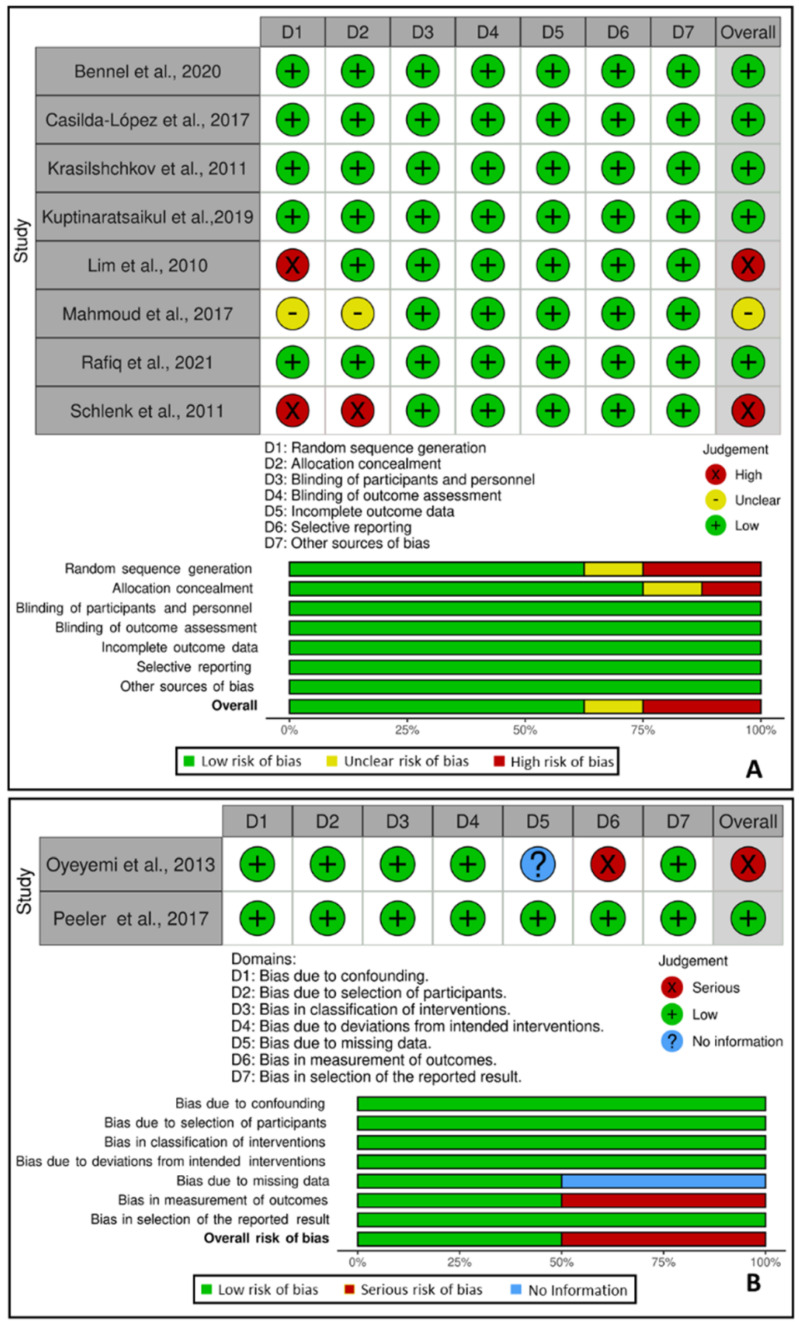
(**A**)—Risk of bias of the randomized controlled trials; (**B**)—Risk of bias of the non-randomized trials.

**Table 1 biology-11-01391-t001:** Publications selected in this systematic review.

Author/Year	Aim	Participants/ Groups/ Mean Age	BMI (kg/m^2^–Mean ± SD)	Physical Exercise Program	Functional Capacity Assessment	PEDro Scale	Results (PF)
Bennel et al., 2020	To directly compare the effects of two exercise programs on primary outcomes of pain and PF	N: 128 Female Age: ≥ 50 years NWB exercise (62.4 ± 6.7) WB exercise (60.9 ± 6.8) Groups: NWB exercise and WB exercise	NWB exercise (37.3 ± 6.8) WB exercise (37.8 ± 6.0)	12 weeks Total of 5 individual sessions 30–40 min for each session Home program: 12 weeks/4 times per week NWB exercises group—Five exercises performed in sitting or supine positions aimed at QS. Three sets of 10 repetitions for each exercise. Resistance applied with an ankle cuff weight or resistance band. WB exercises group—5 exercises performed in a WB position, aimed to strengthen hip abductors, hip extensors, quadriceps and hamstrings	WOMAC KOOS 30-s CSS 40-m FPWT 6-step SCDT	8/10 (high)	Clinically relevant benefits on the primary outcomes of pain (NWB 4.1; WB 3.4) and PF (NWB 21.2; WB 18.8) in both groups over 12 weeks, however, WB exercise may be preferred over NWB exercise
Mahmoud et al., 2017	To evaluate the effects of ITE on quadricep muscle architecture and strength	N: 44 Male Age: 40–65 years IET (54.6 ± 8.6) Control (53.2 ± 9.6) Groups: IET and CG	IET (35 ± 4.1) CG (34.8 ± 4.2)	12 weeks and 3 times per week Exercise: IET (3–5 sets of 5–10 repetitions of 5 s unilateral isometric knee extensions, with 30 s rest between repetitions and 1 min between sets) Both Groups: CPTP (hot packs and therapeutic ultrasonic)	WOMAC	8/10 (high)	↓ WOMAC scores IET Pre: 35.8 ± 4.59 Post: 19.2 ± 7.28 *p* < 0.05
Casilda-López, et al., 2017	To evaluate the effects of a dance-based AEP on functionality, cardiorespiratory capacity, post-exercise heart rate and fatigue in obese postmenopausal women	N: 34 Female Age ≥ 50 years Dance-based AEP (65.62 ± 7.15) Control (66 ± 6.35) Groups: dance-based AEP and CG	Dance-based AEP (31.69 ± 2.44) CG (33.65 ± 3.04)	8 weeks and 3 times per week Exercise: Heated chest-high swimming pool; dance protocol with a 12 min warm-up, followed by 5 min of slow rhythm music, 3 min of fast rhythm music, 5 min slow, 3 min fast and 5 min slow (total 21 min). A 12 min cool-down after the last rhythm CG performed conventional aquatic exercises	WOMAC	8/10 (high)	↓ WOMAC Aggregate score Post-treatment: Dance: 37.30 ± 16.61 CG: 41.83 ± 13.69 *p* = 0.048 Follow-up: Dance: 38.60 ± 13.61 CG: 42.60 ± 9.05 *p* = 0.038
Schlenk, et al., 2011	To examine the effects of a 6-month self-efficacy-based, individually delivered, lower-extremity exercise and fitness walking intervention with a 6-month follow-up on PF	N: 26 Male and Female Age (mean ± SD): 63.2 ± 9.8 years Groups: Staying Active with Arthritis (STAR) group and CG	33.3 ± 6.0	24 weeks 6 weekly sessions + 9 biweekly telephone counseling sessions + HPE lower-extremity flexibility and strengthening; walking toward a goal of 150 min per week + HP CG received usual care and after 6 months received no self-efficacy-based adherence counseling	6-min aerobic endurance walk test SPPB 4-m walk test Standing balance test WOMAC	5/10 (fair)	From baseline to the end of the 6 month follow up: ANOVA HPE: time effect *p* = 0.028 6-min aerobic: ↑ distance *p* = 0.006 SPPB: ↑ scores *p* = 0.002
Kuptinaratsaikul et al., 2019	To investigate the efficacy of a four-week UTE regimen compared to a home exercise regimen relative to pain relief and functional improvement in obese patients with KOA	N: 80 Male and Female Age 50–80 years HPE (61.7 ± 6.9) UTE (62.1 ± 6.4) Groups: Daily quadricep exercise at home HPE N: 40 (CG) or UTE N: 40	HPE (28.4 ±3.0) UTE (28.9 ± 3.2)	4 weeks Study Group: UTE with moderate intensity (NRS 5–6/10) 16 for 30 min, including warm up and cool down, three times per week (12 total sessions) HPE: brochures advising them how to use their knee joints in daily practice + regular quadricep exercise (10–20 repetitions set with a 1–2-min rest) and then to repeat this exercise–rest cycle. Seven days per week/thirty min daily	6MWD QS	8/10 (high)	Improvements in both groups Mean differences (95% CI) 6MWD: 10.81 (−11.90, 3.53) *p* = 0.345 QS: 0.67(−0.10, 1.44) *p* = 0.088
Krasilshchikov et al., 2011	To determine the effect of PCRAE on early primary KOA in overweight and obese	N: 16 Female Age 50–64 PCRAE N: 8 (58.38 ± 4.9) CG N: 8 (58.25 ± 5.1)	PCRAE (28.6 ± 5.6) Non-exercising CG (26.5 ± 4.1)	8 weeks PCRAE training grouped in a single one-hour training session 3 times per week, two sets of isometric quadricep and hamstring contraction, isotonic quadricep contractions, chair squats and dynamic stepping. Resistance: walking for 15 min progressively	WOMAC PF score KECPT right and left 120°/s and 180°/s 6MWD	7/10 (high)	Pretest × Post test PCRAE ↓ WOMAC PF score 21.13 ± 7.9 × 9.50 ± 4.3; *p* < 0.05 KECPT right and left *p* < 0.05 6MWD: 337.65 ± 55.1 × 385.81 ± 32.8 *p* < 0.05
Lim et al., 2010	To design an AEP and LBEP to enhance knee function and reduce body fat in patients with obesity and KOA	N: 75 Male +Female AEP N: 26; Age: 65.7 ± 8.9 LBEP: N: 25; Age: 67.7 ± 7.8 CG: N: 24; Age: 63.3 ± 5.3	AEP (27.9 ± 1.5) LBEP (27.6 ± 1.7) CG: (27.7 ± 2.0)	3 times per week 8 weeks AEP: 40-min duration per session LBEP: 40 min per session, intensity of LBEP began from 40% or 60% of the one repetition maximum for the beginner or advanced classes, respectively	WOMAC PTKE PTKF	7/10 (high)	Before × after ↓ WOMAC PF score AEP:35.1 ± 11.3 × 20.9 ± 9.9 *p* < 0.05 LBEP: 33.6 ± 12.6 × 23.6 ± 12.8 *p* < 0.05 GC: 30.4 ± 19.1 × 27.6 ± 18.3 PTKE and PTKF both with improvement but without statistical significance
Rafiq et al., 2021	To determine the short-term effects of the home protocol of strengthening exercises in NWB positions on the functional capacity of overweight and obese people with KOA	N: 50 Male + Female LLRP: N:25 Age (53.40 ± 5.18) CG: N: 25 Age (52.84 ± 5.74)	LLRP: (32.18 ± 4.49) CG: (32.01 ± 3.89)	3 times per week 4 weeks LLRP: 45–60 min of lower limb resistance training following the IDC in sitting and lying positions CG: whole body ROM exercises following the IDC	WOMAC	8/10 (high)	LLRP and CG: WOMAC PF score *p* > 0.05 (*p* = 0.104)
Peeler et al., 2017	Evaluate the effect of a 12-week LBPP-supported low-load treadmill walking program on knee pain, joint function and performance of daily activities in patients with KOA	N: 35 Age: 50–75 Female:22 63.1 ± 5.9 (53–72) Male: 9 67.1 ± 6.4 (59–75) Single Group	32.8 ± 6.5	12 week-LBPP-supported low-load treadmill walking program. Two times per week for 30 min on a treadmill at a set speed of 3.1 mph at a 0° incline LBPP increased by 5% per min LBPP support was restricted to a maximum of 40% body weight	KOOS COPM Isokinetic muscle strength	Pedro Scale N/A	KOOS (daily living) Female ↑ scores (*p* = 0.0016) 58 ± 15 × 67 ± 15 Male: no statistical difference COPM (performance) Female (*p* = 0.0003) 4.3 ± 1.5 × 5.3 ± 1.6 Male (*p* = 0.0415) 5.0 ± 1.6 × 6.2 ± 2.5 Improvement of Isokinetic muscle strength Female (*p* = 0.0002) 2.9 ± 2.2 × 1.0 ± 1.3 Male (*p* = 0.0490) 4.0 ± 2.0 × 2.8 ± 2.1
Oyeyemi et al.,2013	To investigate the effects of BMI on pain and PF during a four-week exercise program in patients with KOA	N: 46 31 Male 15 Female Age: 34–69 years ± 55 years Groups: normal weight (N: 15) age (56.93 ± 9.56); overweight (N: 13) age (55.31 ± 8.07) obese (N: 18) age (55.2 ± 9.04)	Not mentioned	10 bouts of 10 repetitions of isometric exercise of the quadriceps (10 s of work and 5 s of rest) and 2 min of rest between bouts (total of 100 contractions). Riding the SBE with an initial resistance set at 25 W for 6 min. The resistance was increased to 35, 45 and 55 W in the second, third and fourth week, respectively 2 times per week	WOMAC	Pedro Scale N/A	↓ PF WOMAC scores (week 1 × week 4) Overweight: 40.66 ± 15.6 × 19.81 ± 5.02 Obese: 52.87 ± 12.76 × 19.58 ± 4.56

Abbreviations: AEP: aquatic exercise program; CG: control group; COPM: Canadian occupational performance measure; CPTP: conventional physical therapy program; CSS: chair sit-to-stand; FPWT: fast-paced walk test; HPE: home program exercise; IDC: instructions of daily care; IG: intervention group; ITE: isometric training exercise; KECPT: knee extensor concentric peak torque; KOA: knee osteoarthritis; KOOS: Knee Injury and Osteoarthritis Outcome Score; LBEP: land-based exercise program; LBPP: lower body positive pressure; LLRP: lower limb rehabilitation protocol; N/A: not applicable; NWB: non-weight bearing; PCRAE: progressive and combined resistance and aerobic exercise; PEDro score—(a) ‘high’ methodological quality ≥ 7, (b) ‘fair’ methodological quality = 5 or 6, (c) ‘poor’ methodological quality ≤ 4; PF: physical function; PTKE: peak torque knee extensor; PTKF: peak torque knee flexor; QS: quadricep strengthening; ROM: range of motion; SBE: stationary bicycle ergometer; SCDT: stair-climb; SD: standard deviation; UTE: underwater treadmill exercise; WB: weight bearing; WOMAC: Western Ontario and McMaster Universities Osteoarthritis Index; 6MWD: 6-min walking distance; ↑: increase; ↓: decrease.

## Data Availability

All data were extracted from studies found in the five databases, which were MEDLINE/Pubmed, Physiotherapy Evidence Database (PEDro), Scopus, Cumulative Index to Nursing and Allied Health Literature (CINAHL) and Web of Science.

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
