# Peer review of "Effects of Physical Exercises Alone on the Functional Capacity of Individuals with Obesity and Knee Osteoarthritis: A Systematic Review"

_biology, 2022, doi:10.3390/biology11101391_

Round 1
Reviewer 1 Report
The reviewed article presents a systematic review and meta-analysis of publications on the impact of physical exercise on improving the functional capacity of people with obesity and osteoarthritis of the knee. The authors of the article carefully described the process of searching and selecting scientific publications related to the topic of the work. The literature review covered 674 articles, which were carefully analyzed. Selected articles were used to conduct a meta-analysis. The conducted analysis was thoroughly described and its results were widely discussed. I highly appreciate the quality of the research carried out and recommend the article for publication.
Reviewer 2 Report
Abstract
1) Please revise and edit the type of review as systematic review
2) Conclusion is missed in the abstract
Introduction
1) Please provide the reference for the first sentence of introduction "According to the Osteoarthritis Research Society International (OARSI), osteoarthritis (OA) is a disorder involving mobile joints"
2) Line 44, Do not start the sentence with abbreviation
Additional comments:
Authors have done a systematic review on efficacy of physical exercises alone on the functional capacity of individuals with obesity and knee osteoarthritis. The manuscript is well written, and the study is well conducted. As previously published systematic review and meta-analysis on efficacy of Exercise in Patients with Obesity and Knee Osteoarthritis, authors have systematically reviewed the efficacy of physical exercise alone on obesity and OA with an approximately larger number of included studies. There is a lack of evidence-based analysis, a quantitative study design such as meta-analysis to assess the result would significantly improve the findings.
Title
The word stage one is not very clear in the title to which it refers to.
Reviewer 3 Report
The systematic review submitted by da Silva Caiado and colleagues aims to summarize the effects of PE on the functional capacity of individuals with obesity and KOA and without an associated diet program. The manuscript is very well written, clearly structured and meets the criteria for a systematic review. It is definitelly of interest to the field. However, a major drawback of the study is the lack of published articles with the inclusion criteria of the review. Only 6 of the 10 publications included in the systemtic review show an overall low risk of bias. This makes the conclusions uncertain and raises a question for possible revision of the inclusion criteria. Is it possible to include studies published in different languages?
Minor points:
L86: include are in the sentence: "What ... the effects of physical exercise alone..."
L136: define the abbreviation RCT
L139: studies instead of studied
Author Response
Dear reviewer, please find enclosed our responses. With the best regards, Pr Taiar
